# Comprehensive deletion landscape of CRISPR-Cas9 identifies minimal RNA-guided DNA-binding modules

Arik Shams[1,12], Sean A. Higgins[1,2,3,12], Christof Fellmann [1,4,5], Thomas G. Laughlin [1,6], Benjamin L. Oakes[1,2,3], Rachel Lew[4], Shin Kim[1,2], Maria Lukarska[1,2], Madeline Arnold [1], Brett T. Staahl[1,2,3], Jennifer A. Doudna [1,2,4,7,8,9,10,11] & David F. Savage [1,2✉]

Proteins evolve through the modular rearrangement of elements known as domains. Extant, multidomain proteins are hypothesized to be the result of domain accretion, but there has been limited experimental validation of this idea. Here, we introduce a technique for genetic minimization by iterative size-exclusion and recombination (MISER) for comprehensively making all possible deletions of a protein. Using MISER, we generate a deletion landscape for the CRISPR protein Cas9. We find that the catalytically-dead *Streptococcus pyogenes* Cas9 can tolerate large single deletions in the REC2, REC3, HNH, and RuvC domains, while still functioning in vitro and in vivo, and that these deletions can be stacked together to engineer minimal, DNA-binding effector proteins. In total, our results demonstrate that extant proteins retain significant modularity from the accretion process and, as genetic size is a major limitation for viral delivery systems, establish a general technique to improve genome editing and gene therapy-based therapeutics.

[1] Department of Molecular and Cell Biology, University of California, Berkeley, Berkeley, CA 94720, USA. [2] Innovative Genomics Institute, University of California, Berkeley, Berkeley, CA 94720, USA. [3] Scribe Therapeutics, Alameda, CA 94501, USA. [4] Gladstone Institutes, San Francisco, CA 94158, USA. [5] Department of Cellular and Molecular Pharmacology, University of California, San Francisco, San Francisco, CA 94158, USA. [6] Division of Biological Sciences, University of California, San Diego, San Diego, CA 92093, USA. [7] Graduate Group in Biophysics, University of California, Berkeley, Berkeley, CA 94720, USA. [8] Department of Bioengineering, University of California, Berkeley, Berkeley, CA 94720, USA. [9] Howard Hughes Medical Institute, University of California, Berkeley, Berkeley, CA 94720, USA. [10] Molecular Biophysics and Integrated Bioimaging Division, Lawrence Berkeley National Laboratory, Berkeley, CA 94720, USA. [11] Department of Chemistry, University of California, Berkeley, Berkeley, CA 94720, USA. [12] These authors contributed equally: Arik Shams, Sean A. Higgins. ✉email: savage@berkeley.edu

Domains are the fundamental unit of protein structure[1–3]. Domains are also the unit of evolution in proteins, accumulating incremental mutations that change their function and stability, as well as being recombined within genomes to create new proteins via insertions, fusions, or deletions[4–7]. Extant multi-domain proteins are thus thought to have evolved via the continuous accretion of domains to gain new function[4,8,9]. In addition, eukaryotic proteome diversity is vastly increased by alternative splicing, which tends to insert or delete protein domains[10]. The phenomenon of domain modularity in proteins has been exploited synthetically to rearrange and expand the architecture of a protein, enabling new functionality[11–13]. For example, the programmable DNA nuclease Cas9 can be converted into a ligand-dependent allosteric switch using advanced molecular cloning, similar to other domain insertions dictated by allostery[13,14]. Although there are several methods for comprehensively altering protein topology[15,16], no method has been demonstrated for domain deletion.

Rationally constructed protein deletions have long been essential for elucidating functional and biochemical properties, but are generally limited to a handful of truncations. Moreover, protein engineering can make use of deletions to alter enzyme-substrate specificity[17], enable screens for improved activity and thermostability[18], or minimize protein size[19]. Early approaches to protein deletion libraries resulted in the deletion of single amino acids using an engineered transposon[20,21]. Other methods utilize direct polymerase chain reaction (PCR)[22], random nuclease digestion[23], or random in vitro transposition followed by a complicated cloning scheme[24] to achieve deletion libraries containing a variety of lengths and reading frames. These techniques are low in throughput and/or require complex molecular techniques that poorly capture library diversity; in contrast to protein insertions where library size grows linearly with target length, deletion libraries grow as the square.

A simple and efficient method for building protein deletions coupled with a selection strategy would provide the ability to comprehensively query and delineate the function of domains or motifs in complex and multi-domain proteins. Such a technique could be used to identify crucial functions within multi-domain proteins or splicing variants in a manner akin to how deep mutational scanning can be used to identify the effects of single-nucleotide polymorphisms on functionality[25]. Moreover, with sufficient modularity, the evolutionary path of domain accretion could be explored through iterative combining, or "stacking," of domain deletions to isolate a minimal, core protein for a defined function[7–9].

One attractive target for such a strategy is *Streptococcus pyogenes* Cas9 (SpCas9), the prototypal RNA-guided DNA endonuclease used for genome editing[26]. SpCas9 is an excellent model protein for a comprehensive deletion study because of its multi-domain architecture and availability of high-throughput assays for either DNA cutting or binding[27]. Functionally, SpCas9 targets and cleaves DNA in a multistep process. First, an apo Cas9 molecule forms a complex with a guide RNA (gRNA), containing a 19–22 bp variable "spacer" sequence that is complementary to a DNA target locus. The primed ribonucleoprotein (RNP) complex then surveils genomic DNA for a protospacer-adjacent motif (PAM)—5′-NGG-3′ in the case of SpCas9, where N is any nucleobase—that initiates a transient interaction with the protein to search for an adjacent ~20-bp target sequence. If a target is present, the double-stranded DNA (dsDNA) helix is unwound, allowing the gRNA to anneal to the DNA and form a stable RNA–DNA hybrid structure called an R-loop (see illustration in Supplementary Fig. 8). The formation of a complete 20-bp R-loop triggers a conformational change in Cas9 to form the catalytically active complex[28–30].

SpCas9 has a bi-lobed architecture consisting of the RECognition lobe, responsible for recognizing and binding DNA sequences, and the NUClease lobe, which possesses HNH and RuvC domains that cut the target and nontarget strands of DNA, respectively. Cas9 is postulated to have evolved via domain accretion from a progenitor RuvC domain[9,31]. Consequently, Cas9 orthologs possess manifold architectures. For example, the SpCas9 REC lobe possesses three domains (REC1, REC2, and REC3), while the *Staphylococcus aureus* Cas9 (SaCas9) has a contiguous REC domain without REC2[32,33]. The function of REC2 is ambiguous, but is thought to act as a conformational switch to trigger DNA cleavage[34,35], raising the question of how SaCas9 accomplishes the effect[36]. Thus, the multi-domain, multifunctional nature of Cas9s makes them an excellent model system for exploring domain deletions. Relatedly, Cas9's large size also complicates its delivery using viral vectors. Knowledge of functional deletions may thus facilitate the delivery of genome-editing therapeutics.

Here, we introduce genetic minimization by iterative size-exclusion and recombination (MISER), a technique for systematically exploring in-frame deletions within a protein. Application of MISER to the catalytically dead SpCas9 (dCas9) identified regions of the protein that can be deleted with minimal consequence to binding function. Furthermore, we stacked individual deletions to engineer clustered regularly interspaced short palindromic repeats (CRISPR) effector (CE) proteins that are <1000 amino acids in length. CRISPR interference (CRISPRi) and biochemical assays demonstrated that these variants remain competent for target DNA binding, but are less functional than single-deletion variants. Finally, to understand the structural consequence of deletion, we used single-particle cryo-electron microscopy to solve a 6.2 Å structure of the smallest, 874 amino acid CE. This structure surprisingly revealed an overall conformation that preserves essential functions of SpCas9—emphasizing the concept of domains as independent modules—even though the quaternary structure is severely modified.

## Results

**MISER reveals the comprehensive deletion landscape of SpCas9.** The general concept of MISER is to create a pool of all possible contiguous deletions of a protein and analyze them in a high-throughput fitness assay. The process can then be iterated to stack deletions together. We created such a library by (i) systematically introducing two distinct restriction enzyme sites, each once, across a gene on an episomal plasmid, (ii) excising the intervening sequence using the restriction enzymes, and (iii) re-ligating the resulting fragments (Fig. 1A). In the instantiation here, two separate restriction enzymes (*Nhe*I and *Spe*I) with compatible sticky ends are used. Cleavage, removal of intervening sequence, and ligation thus result in a two-codon scar site (encoding either Ala-Ser or Thr-Ser) not recognized by either enzyme, thereby increasing the efficiency of cloning and enabling iteration of the entire process (Supplementary Fig. 1).

The MISER library was made for nuclease dCas9 as follows. First, single *Nhe*I or *Spe*I sites were systematically introduced into a dCas9 gene with flanking *Bsa*I sites using a targeted oligonucleotide library and recombineering (Supplementary Fig. 1)[37,38]. Second, these plasmid libraries were isolated, digested, respectively, with *Bsa*I and either *Nhe*I or *Spe*I, and then ligated together (Supplementary Fig. 1B). The resulting ligation of gene fragments produces deletions, as well as duplications, such that a MISER library has a triangular distribution, with near-wild-type (WT) length proteins most frequent and the largest deletions least frequent (Fig. 1C). To empirically determine the size range of functional deletions, the dCas9-MISER library was separated on

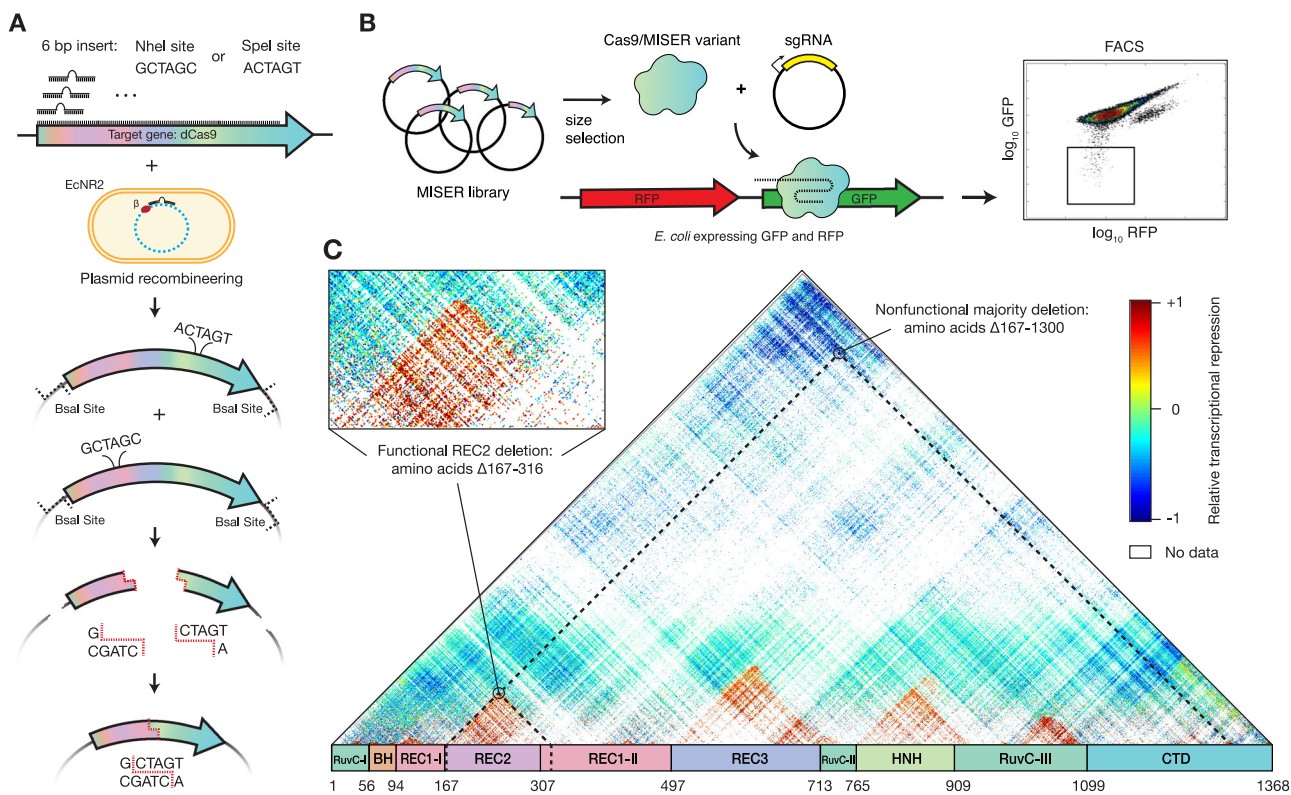

**Fig. 1 Minimization by iterative size exclusion and recombination (MISER). A** MISER library construction. A 6-bp *Spe*I or *Nhe*I recognition site is inserted separately into a dCas9-encoding plasmid flanked by *Bsa*I sites using plasmid recombineering. The resultant libraries are digested with *Bsa*I and either *Spe*I or *Nhe*I, and the two fragment pools are combined and ligated together to generate a library of dCas9 ORFs possessing all possible deletions. **B** The MISER library is cloned into a vector and co-transformed in *E. coli* expressing RFP and GFP with a sgRNA targeting GFP. The library products are expressed, functional variants bind to the target, and repress the fluorophore. Repression activity in vivo is measured by flow cytometry. **C** Enrichment map of the MISER deletion landscape of *S. pyogenes* dCas9. A single pixel within the map represents an individual variant that contains a deletion beginning where it intersects with the horizontal axis moving to the left (N) and ends where it intersects with the axis moving to the right (C). Larger deletions are at the top, with some deletions almost spanning the whole protein. The heatmap shows relative repression activity of variants from two FACS sorts of a single replicate. The map is a composite of Slice 4 and Slice 5 in Supplementary Fig. 3A, B, which present variant ratios post- versus pre-FACS sorting.

an agarose gel and divided into six sublibrary slices of increasing deletion size. The sublibraries were then independently cloned into expression vectors and assayed for bacterial CRISPRi green fluorescent protein (GFP) repression via flow cytometry (Fig. 1B and Supplementary Fig. 2)[39,40]. Sublibrary Slice 4 (ranging from 3.2 to 3.5 kb) was the most stringent (i.e., smallest) library with detectable repression, and functional variants became more frequent in slices possessing smaller deletions, as expected.

Fluorescence-activated cell sorting (FACS) and sequencing of MISER variants identified dCas9 deletion variants competent for DNA binding. To focus sequencing on functional variants, Slice 4 and Slice 5 were sequenced pre- and post-FACS sorting, and the enrichment or depletion of individual variants was quantified (Supplementary Fig. 3). Four large deletion regions were independently identified in both libraries. Although the libraries target different size ranges, their overlapping data were significantly correlated (Supplementary Fig. 3). These data were normalized and combined to generate a comprehensive landscape of functional dCas9 deletions (Fig. 1C). Eighty percent of sequencing depth was focused on deletions from 150 to 350 amino acids in length (Slice 4), and 51.4% (115,530/224,718) of these deletions were detected. Overall, this landscape includes data for 27.5% of all possible dCas9 deletions (257,737/936,396). The four large deletion regions roughly corresponded to the REC2, REC3, HNH, and RuvC-III domains. While larger deletions are bounded between domain termini, small deletions and insertions (~10 amino acids) are tolerated in much of

the structure (Supplementary Fig. 4), a finding that has been previously observed in other proteins[17,22]. Two clear exceptions are the mechanistically essential "bridge helix"[35], which orders and stabilizes the R-loop[41,42], and the "phosphate lock loop"[43], which interacts with the PAM-proximal target strand phosphate to enable gRNA strand invasion. It should be noted that the enrichment data presented here is somewhat sparse and only a relative measurement of CRISPRi function; the larger-scale features of acceptable domain and sub-domain level deletions were, therefore, extensively validated with further in vivo and biochemical assays.

**Cas9 tolerates large deletions while retaining DNA-binding function.** To validate the deletion profile, individual variants from each of the four large deletion regions were either isolated from the library (Supplementary Fig. 5) or constructed via PCR and assayed individually. Representative variants from these regions could be identified that exhibited bacterial CRISPRi nearly as effectively as full-length dCas9 (Fig. 2A and Supplementary Fig. 5). Intriguingly, there are regions within our identified deletions that have been previously tested based on rational design, providing additional insight into the biochemical mechanisms lost with the removal of each domain[35,44]. The most obvious of the acceptable deletions are of the HNH domain that is responsible for cleaving the target strand and gating cleavage by the RuvC domain; it was thus of little surprise that deletions of HNH were tolerated in a molecule that is required to bind but not

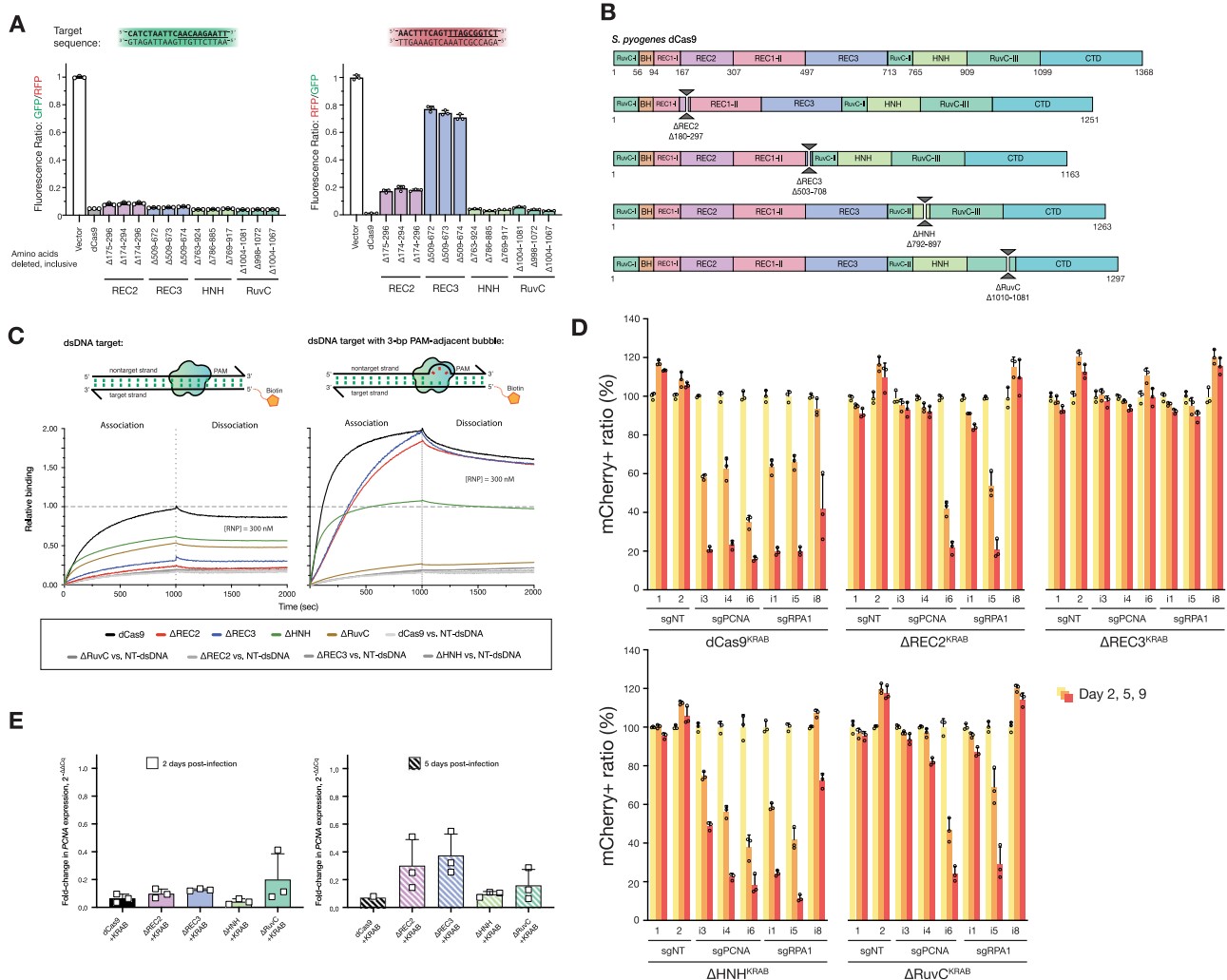

**Fig. 2 Cas9 tolerates whole-domain deletions while maintaining target-binding activity. A** In vivo transcription repression activity of MISER-dCas9 variants with specified amino acids deleted, targeting either GFP (left) or RFP (right). dCas9s with REC2, REC3, HNH, or RuvC domain deletions have near-WT binding activity when targeted to GFP. When targeted to RFP, ΔREC2, and ΔREC3 show less robust binding activity. Data are normalized to vector-only control representing maximum fluorescence. Data are plotted as mean ± SD from biological triplicates. **B** Schema showing cloned MISER constructs with individual domain deletions corresponding to tolerated regions found in MISER screen. **C** Bio-layer interferometry (BLI) assay of MISER constructs. ΔREC2 and ΔREC3 exhibit weak binding against a fully complementary dsDNA target, while ΔHNH and ΔRuvC show intermediate binding. Binding is rescued to near-WT levels in ΔREC2 and ΔREC3, although at a slower rate, when the dsDNA contains a 3-bp bubble in the PAM-proximal seed region. Data are normalized to dCas9 binding to fully complementary dsDNA. **D** U-251 cells stably expressing the indicated MISER-dCas9 or WT-dCas9 KRAB fusion. Proteins were transduced with mCherry-tagged lentiviral vectors expressing sgRNAs targeting essential genes (sgPCNA, sgRPA1) or nontargeting controls (sgNT). At Day 2 post transduction, cells were mixed with the respective parental population; mCherry fluorescence was monitored over time. Data represent the mean and SD of triplicates ($n = 3$). Significance in cell depletion was assessed by comparing samples to their respective Day 2 controls using unpaired, two-tailed $t$ tests ($\alpha = 0.01$). **E** Measurement of CRISPRi efficacy of single-deletion MISER constructs in mammalian U-251 cells using RT-qPCR. U-251 cells were stably transduced with lentiviral vectors encoding dCas9 or MISER constructs fused with a KRAB repressor, along with lentivirus expressing sgRNA targeting *PCNA*. Cells were harvested 2 (left panel) or 5 (right panel) days post transduction of the sgRNA and assayed for PCNA expression. Bar graphs represent fold change of PCNA expression relative to a nontargeting sgRNA. Data presented as mean and SD (for triplicates where shown). Source data are provided as a Source Data file.

cleave DNA. In fact, Sternberg et al. previously demonstrated that an HNH-deleted (Δ768–919) Cas9 is competent for nearly WT levels of binding activity, but is unable to cleave[45]. In contrast, we also uncovered a deletion in the RuvC-III domain that has never been observed. Modeling this deletion on the previously determined structure of SpCas9 bound to a DNA target (PDB ID 5Y36)[46] revealed that it removes a large set of loops, an alpha helix and two antiparallel beta sheets (Supplementary Fig. 7). This deletion does not seem to overlay with a known functional domain and thus may serve as a module that further stabilizes the RuvC domain as a whole. In addition, this deletion abuts the

nontarget and target strand DNA (distance of ~4–6 Å) and may provide a highly useful site to replace with accessory fusions, such as deaminases suitable for base editing the nontarget strand, as was engineered with circularly permuted base editors[16,47].

Our observations for the REC2 and REC3 domains likewise expand upon two rationally engineered deletions. Chen et al. previously demonstrated that the REC3 domain gates the rearrangement of the HNH cleavage by sensing the extended RNA:DNA duplex[44]. Deletion of this domain (Δ497–713) ablated cleavage activity while maintaining full binding affinity. Nishimasu et al. also previously deleted the REC2 domain because they

postulated that it was unnecessary for DNA cleavage, as it is poorly conserved across other Cas9 sequences and lacks significant contact to the bound guide:target heteroduplex in the structure; however, the deletion mutant was found to have reduced activity[35].

To further validate the function and potential deficits of these single whole-domain deletions, we biochemically analyzed representative deletions of each of the REC2, REC3, HNH, and RuvC domains (Fig. 2B). These single-deletion constructs are henceforth referred to as ΔREC2 (residues 180–297 deleted), ΔREC3 (Δ503–708), ΔHNH (Δ792–897), and ΔRuvC (Δ1010–1081). Deletion variants were expressed, purified (see Supporting information and Supplementary Fig. 9 for purification data), and assayed for DNA-binding activity using bio-layer interferometry (BLI) (Fig. 2C and Supplementary Fig. 10). Binding assays revealed that the REC2 deletion confers a defect in binding to a fully complementary dsDNA when complexed with a single-guide RNA (sgRNA). Interestingly, the defect is almost fully rescued upon the addition of a 3-bp mismatch bubble between the target and nontarget DNA strands adjacent to the PAM. DNA unwinding is initiated by Cas9 at the PAM-adjacent seed region, enabling the RNA–DNA R-loop hybrid to form. Rescue via seed bubble, therefore, suggests a potential role for the REC2 domain in unwinding dsDNA.

A similar phenomenon is observed with the ΔREC3 variant, although the binding defect is less pronounced than in ΔREC2. ΔREC3 is also unable to bind fully complementary dsDNA—an effect that is rescued by the same PAM-adjacent 3-bp bubble in the dsDNA substrate, implying a similar DNA unwinding function by the REC3 domain. These results suggest that both the REC2 and REC3 domains are not essential for DNA binding by SpCas9, but may have evolved as "enhancer" domains to allow SpCas9 to more efficiently bind DNA inside the cell.

When measuring the repression activity of the ΔREC3 constructs in vivo, we also observed that the ΔREC3 appears to exhibit varying levels of repression between different gRNA sequences. Specifically, we found that a GFP-targeting gRNA repressed stronger than a red fluorescent protein (RFP)-targeting gRNA with ΔREC3, after controlling for cell growth and fluorophore maturity (Fig. 2A). This was unexpected since the binding of WT Cas9 is generally thought to be gRNA sequence agnostic[48]. One possibility is that the GC content of the targets in GFP and RFP could affect function, for example, a higher proportion of GC base pairing in the "seed" region of a DNA target could present a greater energetic cost of unwinding to a deletion variant like ΔREC3[49]. Analysis of 16 additional spacer sequences and their repression activity relative to WT suggests that this mechanism only moderately ($R^2 = 0.2$) explains the variance (Supplementary Fig. 8). Further comprehensive analysis of the sequence-dependent variability is required to identify the precise energetic threshold the ΔREC3 construct overcomes to unwind DNA.

Similar binding experiments with ΔHNH and ΔRuvC showed that they possess activity intermediate to WT-dCas9 and ΔREC2 or ΔREC3 (Fig. 2C). Surprisingly, adding a 3-bp mismatch bubble adjacent to the PAM does not seem to fully restore the binding function. ΔHNH reaches ~50% binding upon addition of the bubble, performing worse than the ΔREC2 and ΔREC3 constructs upon addition of the bubble. The bubble also does not appear to increase ΔRuvC's binding to dsDNA (Fig. 2C). We speculate that the defect in binding may be due to the R-loop being destabilized by nuclease domain deletion but is stable enough for bulk repression of a fluorophore in culture.

To test whether the MISER constructs retain DNA-binding activity in mammalian systems, we performed CRISPRi to knockdown genes in a U-251 glioblastoma cell line. We transduced target cells with lentiviral vectors expressing our single-deletion MISER constructs (ΔREC2, ΔREC3, ΔHNH, and ΔRuvC) fused to the KRAB repressor domain, followed by selection on puromycin. Stable cell lines were then transduced with a secondary lentiviral vector expressing mCherry fluorescent protein and either a sgRNA targeting one of the essential genes PCNA (sgPCNA) or RPA1 (sgRPA1) or a control nontargeting sgRNA (sgNT). Transduced cells were mixed with the parental populations and monitored for mCherry fluorescence by flow cytometry over several days. We observed that for dCas9 and three of the four single-deletion constructs (ΔREC2, ΔHNH, and ΔRuvC), mCherry fluorescence is markedly lower at 5 and 9 days post transduction, with multiple guides targeting PCNA and RPA1 (Fig. 2D). This suggested that the MISER-expressing mCherry-positive cell lines were repressing essential genes and were depleted from the population. The ΔREC3 construct exhibited little depletion, which is consistent with the BLI data (Fig. 2C) showing that ΔREC3 appears to have a lower association compared to dCas9 and ΔREC2. Western blot data show that the ΔREC3 is expressed at similar levels to the other single-deletion constructs (Supplementary Fig. 11E), so it is unclear why this defect is observed in mammalian cells compared to bacterial repression (Fig. 2A). One possible explanation could be that the mammalian genome is packaged much differently from the bacterial genome, and DNA-targeting proteins have more difficulty accessing heterochromatin.

As the competition assay does not directly measure repression, reverse-transcription quantitative PCR (RT-qPCR) was used to quantitate the expression of PCNA 2 and 5 days post transduction. (Supplementary Fig. 11). RT-qPCR of PCNA showed that after 2 days the ΔREC2, ΔREC3, ΔHNH, and ΔRuvC constructs repress PCNA expression relative to a nontargeting gRNA (sgNT) (Fig. 2E; all measurements are averages ± SD from biological triplicates were shown), with a mean fold change of 0.10 ± 0.03, 0.13 ± 0.01, 0.04 ± 0.03, and 0.20 ± 0.19 in PCNA expression, respectively. At 5 days post transfection, ΔREC2 and ΔREC3 appear to lose some repression activity (0.30 ± 0.20 and 0.38 ± 0.15 fold change relative to sgNT, respectively), while the ΔHNH and ΔRuvC constructs are comparable to WT-dCas9 at Day 5 (0.10 ± 0.020 and 0.16 ± 0.12, respectively) (see Supporting information and Supplementary Fig. 11 for more details on RT-qPCR). Thus, it appears that ΔREC3-KRAB is functional, but does not repress enough to generate a phenotype in our competition assay.

**Stacking MISER deletions results in minimal DNA-binding proteins.** Protein domains are accreted during the evolution of large proteins[3,4,50]. In principle, accretion could be experimentally reversed provided sufficient modularity is present to offset evolutionary divergence, epistasis, and other deleterious effects in "stacked" deletions. To emulate this process, while also engineering a minimal Cas9-derived DNA-binding protein, we generated a library of constructs that consolidated the ΔREC2, ΔREC3, ΔHNH, and ΔRuvC deletions found by the MISER screen.

A library of multi-deletion variants, termed CEs due to their highly pared-down sequence relative to WT Cas9, was constructed as follows: individual sublibraries of deletions from REC2, REC3, and the HNH domains were isolated from the full MISER library. This was done by selecting against the full-length dCas9 sequence by targeting a pre-existing restriction site within each deleted region so that only transformations of circular plasmids that had the respective deletion would be favored (Fig. 3B and Supplementary Fig. 5). The RuvC deletion was an exception since it did not have a pre-existing restriction site; therefore, a manually constructed ΔRuvC variant (Δ1010–1081) was amplified and used as a starting point for further stacking.

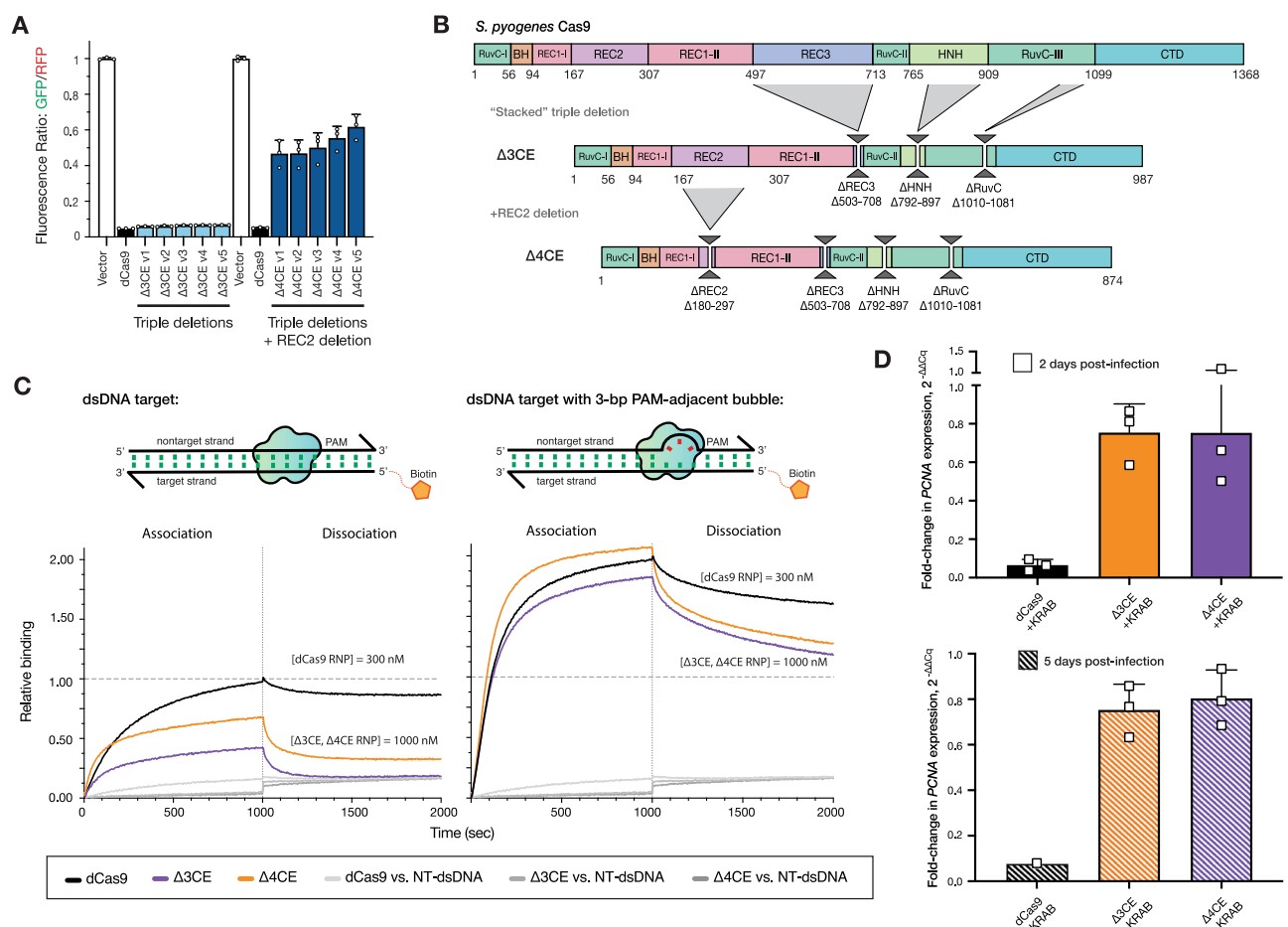

**Fig. 3 Stacking multiple domain deletions on Cas9 results in defective DNA-binding activity. A** In vivo transcription repression activity of MISER CRISPR effectors containing triple (Δ3CE) and quadruple (Δ4CE) deletion variants. Sublibraries of REC2, REC3, HNH, and RuvC were combined to build a library of stacked deletions, and the resulting library was assayed for high-performing variants using FACS (light blue bars). As none of the variants contained a REC2 deletion (~Δ167–307), we named the highest-performing triple-deletion variant in this library (Library 2; see Supplementary Fig. 6) Δ3CE. To force a library containing REC2 deletions, a sublibrary of REC2 deletions was added to Δ3CE, resulting in a library of quadruple deletion variants that contain Δ3CE and a REC2 deletion (dark blue bars). Data are plotted as mean ± SD from biological triplicates. **B** Expression constructs for Δ3CE and Δ4CE, with specified deletions manually cloned in. **C** BLI assay of CE constructs. Δ3CE and Δ4CE exhibit almost no binding against a fully complementary dsDNA target at 300 nM RNP (see Supplementary Fig. 10); and weak binding at 1000 nM RNP. Binding is rescued to near-WT levels when RNP concentration is 3.3× that of dCas9 if the dsDNA contains a 3-bp bubble in the PAM-proximal seed region. Data are normalized to 300 nM dCas9 binding to fully complementary dsDNA. **D** Measurement of CRISPRi efficacy of Δ3CE and Δ4CE in U-251 cells using RT-qPCR. Fold change in PCNA expression levels is measured by RT-qPCR, 2 and 5 days after KRAB-Δ3CE and KRAB-Δ4CE expressing cell lines are transduced with a sgRNA targeting PCNA. Δ3CE and Δ4CE exhibit weak DNA binding and transcriptional repression activity compared to dCas9. Bars represent the fold change of PCNA expression relative to a nontargeting sgRNA. Data are presented as mean and SD (for triplicates where shown). Source data are provided as a Source Data file.

The dCas9 gene was divided into four fragments spanning the major deletions and recombined using Golden Gate cloning (Supplementary Fig. 6). The resulting library, CE Library 1, was assayed using bacterial CRISPRi, and functional variants were isolated by FACS, as above. A variety of functional CEs were obtained (Fig. 3A), although, surprisingly, none of them possessed a REC2 deletion. We, therefore, generated a second library, CE Library 2, in which a library of triple-deletion variants was crossbred with REC2 deletion variants to bias towards a deletion from this region (Supplementary Fig. 6). Again, the most functional CE variants isolated by FACS did not contain REC2 deletions. Finally, in an attempt to force a minimal CE, the most active CE variant from CE Library 1 and 2, termed Δ3CE, was directly combined with a library of REC2 deletions and screened for activity. The resulting "hard-coded" quadruple deletion CE variants all exhibited loss of function relative to WT (Fig. 3A), which explains why the REC2 deletion

was lost in our functional variants. The most active variant (Δ4CE) possessed a deletion of Δ180–297 and was confirmed upon re-transformation to display ~50% activity of WT-dCas9 (Fig. 3A, C) in *Escherichia coli*.

To validate the stacked deletion constructs biochemically, we expressed and purified the Δ3CE and Δ4CE variants from *E. coli* (Fig. 3B and Supplementary Fig. 10). BLI experiments revealed that compared to the bacterial in vivo repression data, the DNA-binding abilities of both stacked deletion constructs were attenuated relative to dCas9 (Fig. 3C). To obtain reasonable kinetic profiles, the concentration of RNP for Δ3CE and Δ4CE was increased to 1000 nM, but even under these conditions both variants lag WT-dCas9 at 300 nM. The PAM-interrogation ability of the two constructs appeared to be intact, as evidenced by the sharp drop-off in signal during the dissociation phase, but both Δ3CE and Δ4CE dissociated at a much higher rate compared to dCas9. The $k_{on}$ was restored upon the addition of a

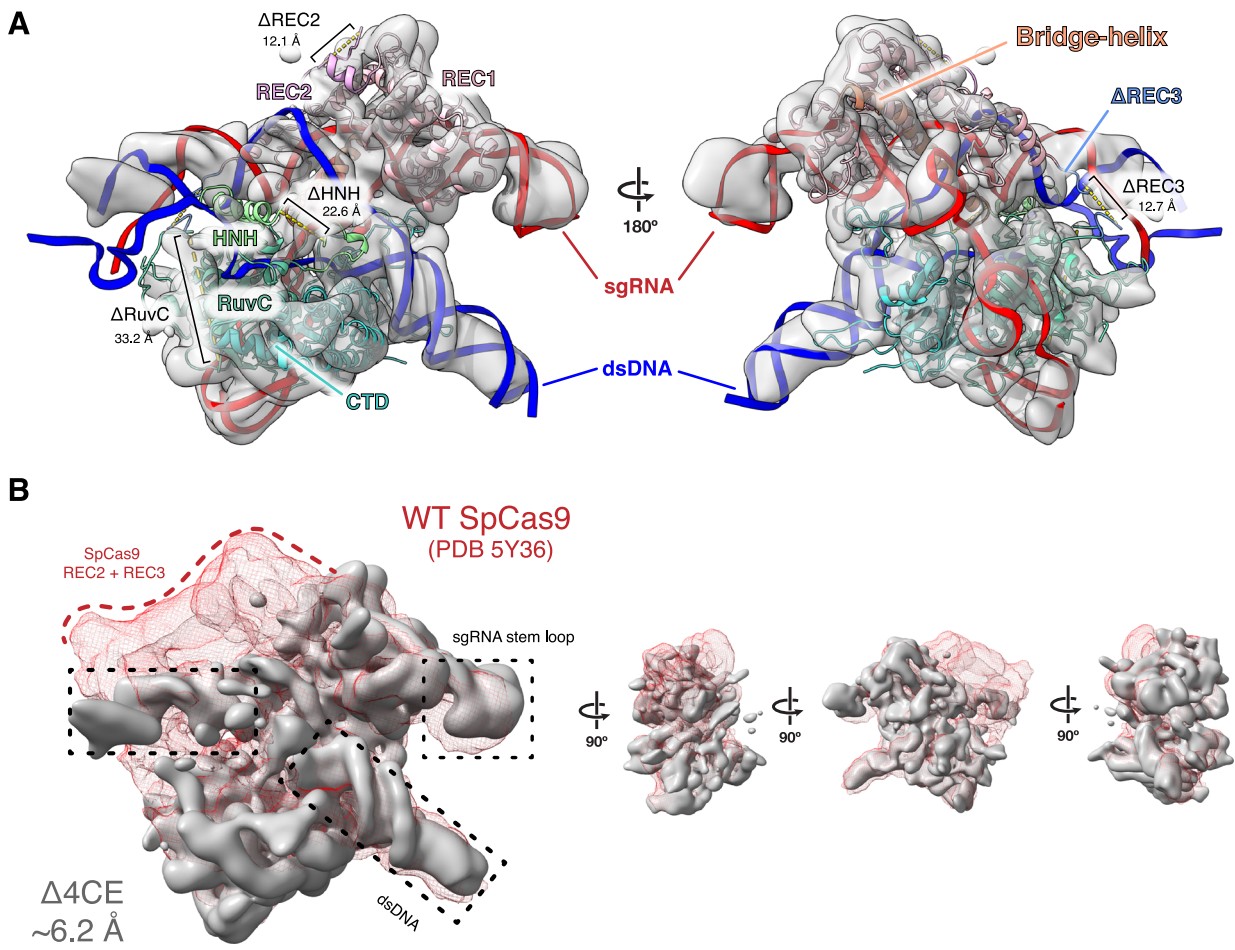

**Fig. 4 Density map of Δ4CE compared to WT SpCas9. A** Single-particle cryo-electron microscopy was used to obtain a density map of the dsDNA-bound RNP complex of the Δ4CE construct at an overall resolution of 6.2 Å (EMD-22518). The light gray volume shows the Δ4CE density overlaid onto RNA–DNA hybrid R-loop (red and blue) and structure of WT SpCas9 (PDB 5Y36). The cartoon model corresponds to the WT SpCas9 structure, showing only the remaining residues and corresponding domains after the REC2, REC3, HNH, and RuvC deletions from the Δ4CE construct are manually removed from the model. Deletion termini are labeled with the distances between termini. **B** Density of Δ4CE cryo-EM overlaid with the WT SpCas9 clearly shows volumes representing dsDNA target and the sgRNA stem loop (black boxes). The red mesh represents the total WT SpCas9 density from EMD-8236.

3-bp bubble, suggesting that these minimal Cas9s possess the kinetic defect in dsDNA binding inherent to both ΔREC2 and ΔREC3. The fact that these minimal constructs are still able to bind DNA in a sgRNA-targeted fashion is surprising, considering that the Δ3CE and Δ4CE constructs retain only ~72% and ~63%, respectively, of the original dCas9 protein primary sequence (Fig. 3B).

We assessed the DNA-binding activity of the CE constructs in mammalian cells similarly to the single-deletion variants described earlier. As before, we performed CRISPRi against PCNA in U-251 cells, this time transducing the Δ3CE and Δ4CE KRAB fusions and sgRNA, followed by mixing with the parental cells and monitoring for mCherry fluorescence for up to 9 days. As expected from the minimal repression in bacteria, we did not observe functional depletion in the competition assay (Supplementary Fig. 11H). We followed the fluorescence assay in mammalian cells with RT-qPCR 2 and 5 days post transfection. Unlike the single-deletion variants, Δ3CE and Δ4CE do not repress nearly as well as dCas9, exhibiting a fold change in PCNA expression relative to nontargeting sgRNA of $0.75 \pm 0.1$ and $0.80 \pm 0.13$, respectively, after 5 days post transduction of the sgRNA (Fig. 3D). This result suggests that the Δ3CE and Δ4CE constructs are functional, but severely defective in DNA binding in a mammalian system.

**The minimal Δ4CE has a similar structure to WT SpCas9**. To understand the structural rearrangement accompanying domain deletion, we used single-particle cryo-EM to determine a reconstruction of the Δ4CE DNA-bound holocomplex (Fig. 4), to a resolution of 6.2 Å (Supplementary Fig. 12). Remarkably, overlaying the density of the Δ4CE construct over the WT SpCas9 R-loop structure (PDB ID 5Y36)[46] as a rigid-body model shows that the minimal complex, consisting primarily of the REC1, RuvC, and C-terminal domains, possesses the same overall architecture as the WT holocomplex (Fig. 4A and Supplementary Fig. 12). The double-helical dsDNA target and the stem loop of the gRNA that are part of the R-loop can be resolved from the density and overlays almost exactly over the WT SpCas9 R-loop. This observation supports the hypothesis that the R-loop is a thermodynamically stable structure that drives the formation of the primed Cas9 RNP–DNA complex[51,52]. Although individual residues cannot be resolved, the remaining RuvC domain in the construct is linked to the C terminus of the REC1 domain via a TS linker (MISER scar), thereby maintaining a bi-lobed complex reminiscent of WT SpCas9. The gRNA-interacting regions of the REC1 and CTD are also spatially conserved, consistent with their observed indispensability on the MISER enrichment map. This raises the question of how the minimal protein is able to form a stable R-loop despite lacking a large part of the REC lobe.

## Discussion

Protein evolution takes large steps through sequence space using domain rearrangements, duplications, and indels[2,53]. While rearrangements, duplications, and insertions have been widely studied, domain deletions are largely under-investigated, due to limited experimental data and the difficulty in properly annotating deletions in protein sequence datasets[54]. Although deletion studies in proteins have been performed, they are limited in their scope regarding the scale of deletions, complexity, and generalizability. In this work, we present a technique that is versatile, comprehensive, and unbiased to probe the deletion landscape of virtually any protein, limited only by the fidelity, and efficiency of a functional screen.

We have used SpCas9 as a proof of concept to demonstrate the utility of MISER because it is a well-characterized, multi-domain protein, easy to assay, and its overall size poses a limit for therapeutic delivery. The WT SpCas9 gene is too large to be packaged into an adeno-associated viral vector (AAV), which has a maximum reported cargo size of <5 kb[55,56] when including the sgRNA sequence and necessary promoters. There are now smaller characterized CRISPR-Cas effectors suitable for AAV delivery by themselves[19,57], but an important need in both research and therapy is delivery of effectors fused to other domains, such as for base-editing and transcriptional activation or repression[58]. MISER may thus find utility in minimizing these much larger constructs. In addition, immunogenicity is emerging as a major issue when developing SpCas9 as a therapeutic and deleting antigenic surface residues can potentially reduce the reactivity of the protein against the immune system[59,60].

We were surprised to discover the effect the deletion of the REC2 domain had on SpCas9 binding. Nishimasu et al. had previously reported that a REC2 deletion (Δ175–307) retained ~50% of editing activity and suggested that the attenuated activity might be due to poor expression or stability[35]. In contrast, our data suggest that the ΔREC2 variant folds and retains target recognition and binding function, but loses DNA unwinding capability. The observation that ΔREC2 binding is restored upon the addition of a 3-bp bubble adjacent to the PAM suggests that the poor binding is due to a kinetic defect. The specific nature of the defect requires further study, although we speculate that the REC2 domain interacts nonspecifically and transiently with the R-loop, perhaps stabilizing the DNA strands during hybridization (i.e., lowering the kinetic barrier) or stabilizing the final R-loop complex (i.e., lowering the energetic cost of unwinding and hybridization)[44].

We also note the observed difference in activity of the MISER constructs between bacterial in vivo repression experiments and the in vitro binding activity using BLI. We speculate that the MISER constructs are inherently defective for binding target DNA, but that sufficiently perturbed dsDNA in bacteria—such as during replication, transcription, or other rearrangements—presents enough opportunity in the form of dynamically un- and under-wound dsDNA, or stretches of single-stranded DNA, to allow the gRNA to anneal to the spacer sequence[52,61]. In addition, abundant or overexpressed proteins, as is the case here, can often achieve concentrations exceeding 1 μM inside E. coli cells, so it is also possible that the overall high abundance of the MISER constructs in the bacterial repression assay is contributing to the binding signal[62].

The effect of the ΔHNH and ΔRuvC deletions was as expected in the bacterial repression assay; however, we were surprised to see that in the in vitro experiments the binding defect was not fully rescued upon addition of the 3-bp bubble in the dsDNA substrate. This suggests that while the REC domains might be conferring a kinetically driven unwinding function to Cas9, the HNH and RuvC nuclease domains might instead have some role

in stabilizing the overall DNA-bound complex. The difference in the in vivo and in vitro conditions may be due to DNA dynamics inside a cell versus in solution.

Finally, in our cryo-EM structure of Δ4CE, we note the remarkable similarity of the protein to WT SpCas9, which underscores the inherent stability of the Cas9 R-loop complex. Previous studies have shown that the formation and maintenance of the R-loop is the molecular "glue" that holds the DNA–RNA–protein complex together[51]. The similitude between the WT and Δ4CE structure also hints at the evolutionary history of SpCas9, suggesting that the "essential" function of the protein was to enable the formation of an R-loop upon a RuvC scaffold for DNA binding and cleavage, which was then tuned by accretion and interactions of other domains—such as those that comprise the REC lobe and the HNH domains[9,63]. Notably, this analysis ignores the role of the gRNA; future iterations of MISER could also be used to evaluate the deletion landscape of CRISPR-associated RNAs.

MISER facilitates the study of protein deletions with unprecedented versatility and efficiency. In this study, we have explored domain modularity and essentiality of CRISPR-Cas9 domains, but MISER can be adapted to any application requiring a reduction in genetic size. AAV-based transgene delivery is subject to a <5 kb payload limit and is a prime target for MISER. Besides CRISPR proteins and their cognate gRNAs, there are numerous other therapeutic proteins limited by their size, such as cystic fibrosis and dystrophin (muscular dystrophy)[55,64]. Beyond threshold effects, even partially reducing the size of AAV genomes can provide a large advantage in packaging efficiency by improving capsid formation[55]. Finally, MISER also reveals small deletions tolerated within proteins, which suggests that this approach could be useful in the development of non-immunogenic biomolecules. Paring away antigenic residues may remove antigenic epitopes on a protein surface, thus allowing the molecule to function without eliciting an immune response[65,66].

## Methods

**Molecular biology**. All restriction enzymes were ordered from New England Biolabs (NEB). PCR was performed using Q5 High-Fidelity DNA Polymerase from NEB. Ligation was performed using T4 DNA Ligase from NEB. Agarose gel extraction was performed using the Zymoclean Gel DNA Recovery Kit, and PCR clean-up was performed using the "DNA Clean & Concentrator," both from Zymo Research. Plasmids were isolated using the QIAprep Spin Miniprep Kit (Qiagen). All DNA-modifying procedures were performed according to the manufacturers' instructions.

**MISER library construction: plasmid recombineering**. Two sets of 1368 oligonucleotides were designed and ordered as Oligonucleotide Library Synthesis from Agilent Technologies (Table S1). Oligonucleotides were designed to insert a six base-pair (bp) recognition sequence for either the restriction enzyme NheI or SpeI between every codon in dCas9 (Supplementary Fig. 1A). The full list of ordered oligonucleotides is available as Auxiliary Supplementary Materials—Recombineering Oligonucleotides. Internal priming sites were included to amplify NheI- or SpeI-specific oligonucleotide libraries. A modified amplification procedure was performed as follows. In a 50 μL PCR reaction, 10 ng of template oligonucleotide library was amplified according to the manufacturer's instructions, but with an extension time of only 5 s, and a total of only 15 cycles. Dimethyl sulfoxide (1.5%) was also included in the PCR reaction. These modifications were empirically determined to minimize undesirable higher-order PCR products that were observed to be produced by amplification. These side products are likely the result of complementary oligonucleotides priming one another. Notably, this phenomenon is likely inherent to the amplification of a library of DNA tiled across a common sequence—in this case dCas9. PCR primers can be found in Table S6 and Auxiliary Supplementary Materials—Primer Sequences. Twenty-four such reactions were typically performed in parallel and then combined, followed by concentration with Zymo DNA Clean & Concentrator. BsmBI restriction digestion was then used to remove priming ends, followed by a second concentration with Zymo DNA Clean & Concentrator, resulting in mature double-stranded recombineering-competent DNA.

Plasmid recombineering was performed as described in Higgins et al.[38], using strain EcNR2 (Addgene, ID: 26931) to generate MISER libraries in plasmid pSAH060. Plasmid sequences can be found in Auxiliary Supplementary Materials

—Plasmid Sequences. Briefly, mature double-stranded recombineering-competent DNA at a final volume of 50 μL of 1 μM, plus 10 ng of pSAH060, was electroporated into 1 mL of induced and washed EcNR2 using a 1 mm electroporation cuvette (Bio-Rad GenePulser). A Harvard Apparatus ECM 630 Electroporation System was used with settings 1800 kV, 200 Ω, 25 μF. Three replicate electroporations were performed, and then individually allowed to recover at 30 °C for 1 h in 1 mL of SOC (Teknova) without antibiotic. LB (Teknova) and kanamycin (Fisher) at 60 μg/mL were then added to 6 mL final volume and grown overnight. A sample of recovered culture was diluted and plated on kanamycin to estimate the total number of transformants, typically >10$^7$. Cultures were miniprepped and combined the next day. Plasmid recombineering is relatively inefficient, and only a fraction of recovered plasmids contained successful *NheI* or *SpeI* insertions. To recover completely penetrant libraries, an intermediate cloning step was performed. A PCR product conferring resistance to chloramphenicol was cloned into both libraries of pSAH060 plasmids (Auxiliary Supplementary Materials—Chloramphenicol Selection). This PCR product contained either flanking *NheI* restriction sites or *SpeI* restriction sites, such that only modified pSAH060 plasmids (possessing *NheI* or *SpeI* restriction sites) could obtain chloramphenicol resistance through *NheI*/*SpeI* digestion and subsequent ligation. Libraries were then purified (Zymo) and transformed into XLI-Blue-competent cells for overnight selection in chloramphenicol (Amresco) at 25 μg/mL, followed by plasmid isolation the next day. Samples of recovered cultures were also plated on both kanamycin alone (native pSAH060 resistance) and chloramphenicol alone (resistance mediated by successful recombineering insertion) to estimate the fraction of modified plasmids and therefore the restriction library size. Recombineering efficiencies were observed at ~0.5% by this method, indicating restriction library sizes of >50,000, well above the number of unique insertion sites per library (1368). Finally, chloramphenicol-resistant pSAH060 libraries were digested with either *NheI* or *SpeI* as appropriate, removing the chloramphenicol cassette. The libraries were run on an agarose gel, and the 5953 bp (5947 bp pSAH060 + 6 bp inserted restriction site) linear band corresponding to each library was gel extracted. To construct deletion variants composed of N- and C-terminal dCas9 fragments, 1 μg of each library was mixed and digested with *BsaI*, then cleaned up (Zymo). The resulting DNA mixture contained equimolar free dCas9 N- and C-terminal fragments, as well as an equimolar pSAH060 vector backbone. This mixture was then ligated in the presence of *SpeI* and *NheI*, "locking" dCas9 fragments together by one of two 6-bp scar sites not recognized by either enzyme (Supplementary Fig. 1B). The ligated MISER library was transformed into XL1-Blue, grown overnight, and plasmids were isolated the next day. The MISER library of dCas9 is quite large, with 936,396 possible deletions ($N$ ($N$ + 1)/2, $N$ = 1368), and all cloning steps were performed with validation that >10$^7$ transformants were obtained.

**MISER library construction: library size selection**. The MISER library is theoretically composed of all possible N- and C-terminal fragments, including both duplications and deletions. To isolate deletions in a particular size range, the MISER library was digested with *BsaI*, to excise the dCas9 gene from the vector backbone and run on an agarose gel. Various slices of the MISER library were individually gel extracted (Supplementary Fig. 2A), ligated into expression vector pSAH063 (Supplementary Fig. 2B), and transformed into *E. coli*.

**Fluorescence repression assays and flow cytometry**. The catalytically dead dCas9-MISER variants were used to repress the transcription of genomically encoded fluorescent reporter genes in *E. coli* as previously described[39]. A sgRNA targeting GFP was transcribed from plasmid pgRNA-bacteria (Addgene, ID 44251)[39], which results in repression of constitutively expressed GFP, contingent on functional dCas9 expression from pSAH063[27]. This repression was quantified relative to non-targeted RFP, which is expressed from the same genomic locus[39]. This assay yields robust repression detection (Supplementary Fig. 2B), with at least an order of magnitude lower GFP signal after 8 h of growth at 37 °C with 750 r.p.m. shaking in LB medi + 1 nM isopropyl β-D-1-thiogalactopyranoside induction of dCas9 from pSAH063. Assays and flow cytometry were conducted in either an M1000 plate reader (Tecan) or an SH800S Cell Sorter (Sony Biotechnology). For GFP/RFP ratiometric measurements (Figs. 2A and 3A), there was no significant difference between samples for the RFP fluorescence measurement.

**Deep sequencing**. One hundred nucleotide single-end reads were used to sequence the dCas9 Slice 4 and Slice 5 libraries. dCas9 open-reading frames were amplified from pSAH064 libraries with primers SAH_356 and SAH_358. PCR products were further prepared for deep sequencing by the UC Berkeley Functional Genomics Laboratory. Sequencing was performed by the UC Berkeley Vincent J. Coates Genomics Sequencing Laboratory on an Illumina HiSeq4000. Samples were mixed at custom ratios as follows: Slice 5 Naive Library: 10%; Slice 5 Sorted Library: 10%; Slice 4 Naive Library: 40%; Slice 4 Sorted Library: 40%. Sequencing reads can be accessed on NCBI GenBank; accession number PRJNA746606. Sequencing analysis was performed with custom MATLAB scripts available online at https://github.com/savagelab. Briefly, reads were analyzed for the novel presence of the two possible MISER scar sequences, "GCTAGT" or "ACTAGC." The majority of reads were fully WT-dCas9 sequences, as expected due to the fact that scar

sequences can occur anywhere along dCas9. Once detected, reads containing 15 bp upstream and downstream of the scar (that exactly matched dCas9 sequence) were used to identify the location of a deletion. Sequencing statistics can be found in Table S3. Enrichment ratios were calculated by taking the ratio of the frequency of each variant before and after selection[67]. To conservatively display variants only detected in one library, one artificial read was added to both datasets. The log base ten of these enrichment ratios were plotted (Supplementary Fig. S3A, B) for each of the two libraries. For visualization, these two datasets were also normalized according to their Pearson's correlation (Supplementary Fig. S3E), combined (the mean was calculated for those variants with two values), and rescaled for display (Fig. 1C and Supplementary Fig. S4A). Variants with large deletions (>1000 bp) as shown in Supplementary Fig. 3C, D are most likely "cheaters," i.e., small plasmids that are missing most of the dCas9 sequence and are therefore more easily replicated and less toxic to the cells.

**In vitro DNA-binding assays**. See Supporting information for detailed protein purification methods. Purified proteins were complexed with 1.2× molar ratio sgRNA in the presence of 5 mM MgCl$_2$. 5′-Biotinylated target DNA and corresponding nontarget DNA was purchased from IDT as single-stranded oligos and annealed 1:1 according to standard IDT protocols. All BLI measurements were performed on an Octet RED384 System (ForteBio). Biosensors coated with streptavidin (SA) were incubated in BLI buffer (20 mM HEPES pH 7.5, 100 mM KCl, 5 mM MgCl$_2$, 10 μg/mL heparin, 50 μg/mL bovine serum albumin, 0.01% (v/v) IGEPAL CA-630, 1 mM tris(2-carboxyethyl)phosphine, 10% (v/v) glycerol) for ~10 min prior to assay. 5′-Biotinylated target DNA (ligand) and corresponding nontarget DNA was purchased from IDT as single-stranded oligos and annealed 1:1 according to standard IDT protocol (see Table S4 for oligo sequences). Biotinylated dsDNA was diluted in BLI buffer to a concentration of 10 nM. dCas9 or MISER construct RNPs were diluted in BLI buffer at various concentrations (0.1× to 10× reported $K_D$). BLI step sequence was as follows: SA biosensors were incubated in BLI buffer for 60 s (baseline); dsDNA ligands were loaded onto SA biosensors for 300 s (loading); SA biosensors were incubated in BLI buffer for 60 s again to re-equilibrate ligand-bound tip (baseline); dsDNA-functionalized biosensors were incubated with RNP analytes for 1000 s (association); and biosensors were incubated in baseline wells from Step 1 for 1000 s (dissociation). All steps were performed at 37 °C with stirring (1000 r.p.m.). Data analysis was performed with Octet Data Analysis HT software (ForteBio).

**Mammalian CRISPRi assay**. For the mammalian CRISPRi-based competitive proliferation assay, human U-251 glioblastoma cells were stably transduced with lentiviral vectors (pSC066) expressing MISER or WT-dCas9 KRAB fusion proteins, followed by selection on puromycin (InvivoGen, #ant-pr-1; 1.0–2.0 μg/mL). The respective cell lines were then transduced with a secondary lentiviral vector (pCF221) expressing mCherry fluorescence protein and either CRISPRi sgRNAs targeting essential genes (sgPCNA, sgRPA1) or nontargeting controls (sgNT). After mixing with the respective parental population (at an ~80:20 ratio of transduced to non-transduced cells), the percentage of mCherry-positive cells was monitored by flow cytometry (Attune NxT flow cytometer, Thermo Fisher Scientific) over several days to assess the effect of CRISPRi with the given Cas9 variant on cell proliferation. CRISPRi sgRNAs had been previously designed[68], as were nontargeting sgRNAs[69]. The sgRNAs were designed with a G preceding the 20-nucleotide guide for better expression from U6 promoters and cloned into the pCF221 lentiviral vector for expression[16]. See Supporting information for details on mammalian cell culture and lentiviral transduction.

**Reverse-transcription quantitative PCR**. To measure the efficacy of CRISPRi repression of essential genes by dCas9-MISER constructs in cultured mammalian cells, we performed RT-qPCR of targeted genes in human U-251 glioblastoma cells. Cells were stably transduced with lentiviral vectors encoding dCas9- or MISER-KRAB proteins, and sgRNA targeting PCNA (sgPCNA-i6) as described in the mammalian CRISPRi experiment (including nontargeting guide sgNT-1), except without any mixing with the parental population. Cells were allowed to grow and then harvested 2 and 5 days post transduction. RNA was extracted using Trizol–chloroform and stored in −80 °C[7]. RNA was reverse-transcribed to complementary DNA (cDNA) with RNA-to-cDNA EcoDry™ Premix with random hexamers (Takara Bio), using the manufacturer's protocols. Quantitative PCR (qPCR) amplification of cDNA was performed using primers specific for PCNA (oAS089-92, Table S4) using SYBR Green PCR Master Mix (Thermo Fisher Scientific) in a QuantStudio 3 Real-time PCR System (Thermo Fisher Scientific). Glyceraldehyde 3-phosphate dehydrogenase (GAPDH) was used as the housekeeping control (amplified with primers oAS117-118, Table S4). All results are reported relative to the expression of PCNA in cells transfected with nontarget gRNA (sgNT-1, Table S4). Only amplification plots below a ΔRn threshold of 0.040 and a $C_t$ value <35 cycles were used for the analysis of expression levels. $\Delta C_q$ values were calculated by subtracting $C_q$ values of GAPDH amplifications from PCNA, and $\Delta\Delta C_q$ values were calculated by subtracting the nontarget samples from the target samples. Fold change in expression is reported as $2^{-\Delta\Delta C_q}$.

**Cryo-electron microscopy sample preparation and image acquisition**. The ternary complex was prepared at 37 °C using a Δ4CE, sgRNA, and dsDNA target at a ratio of 1:1.5:2 in complexing buffer (30 mM Tris-HCl, pH 8.0, 150 KCl, 5 mM MgCl₂, 5 mM dithiothreitol, 2.5 % glycerol). Protein and sgRNA were incubated for 30 min prior to the addition of dsDNA for an additional 1 h of incubation. The sample was then desalted using a spin column (Zeba) into Complexing Buffer containing 0.1% glycerol to be used for grid preparation. To prepare the sample for imaging, 3.2 μL of the ternary complex (around 30 nM) was applied to R1.2/1.3 Cu 200 grids (Quantifoil) coated with a thin layer of homemade continuous carbon that had been glow discharged for 15 s immediately before use. The sample was incubated on the grid at 100% humidity and 16 °C for 10 s prior to blotting for 5 s with filter paper and plunging into liquid ethane cooled to liquid nitrogen temperatures using a Vitrobot Mark IV (TFS). The sample was imaged using a Talos Arctica transmission electron microscope (TFS) operated at 200 kV and equipped with a K3 direct electron detector (Gatan) at the Bay Area Cryo-EM facility at the University of California, Berkeley. Movies were recorded in super-resolution counting mode at an effective pixel size of 0.45 Å, with a cumulative exposure of 60 e⁻·Å⁻² distributed uniformly over 60 frames. Automated data acquisition was performed using image shift and active beam tilt compensation as implemented in SerialEM-v3.7 to acquire movies from a $3 \times 3$ array of holes per stage movement[70]. In total, 3400 movies were acquired with a realized defocus range of −1.5 to −3.8 μm. See Supporting information for details on cryo-EM image processing and modeling.

**Reporting summary**. Further information on research design is available in the Nature Research Reporting Summary linked to this article.

## Data availability

Naive and sorted sequencing reads for Slices 4 and 5 can be accessed from NCBI GenBank; accession code PRJNA746606. Wild-type SpCas9 cryo-EM data were downloaded from the Electron Microscopy Data Bank (EMDB); accession code 8236. The 3D model for the structure was obtained from the Protein Data Bank (PDB); entry 5Y36[46]. All sequencing data that support the findings of this study are available from the authors upon reasonable request. Cryo-EM data for the Δ4CE construct are available at EMDB; accession code 22518. All other relevant data are available from the corresponding author on request. Source data are provided with this paper.

## Code availability

All custom scripts are available at https://github.com/savagelab [https://doi.org/10.5281/zenodo.5098292].

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

## Acknowledgements

This work was supported by NIH grants 1R01GM127463 (to D.F.S.) and RM1HG009490 (to J.A.D. and D.F.S.). Additional support and reagents were provided by Agilent Technologies. A.S. was supported by the NSF GRFP (grant no. 1752814), S.A.H. was supported by NIH Training Grant 5T32GM066698-10, and M.A. was supported by the ARCS Foundation. C.F. was supported by a US NIH K99/R00 Pathway to Independence Award (K99GM118909, R00GM118909) from the NIGMS. J.A.D. is an investigator of the Howard Hughes Medical Institute (HHMI), and this study was supported in part by HHMI. This work used the Vincent J. Coates Genomics Sequencing Laboratory at the University of California, Berkeley, supported by NIH S10 instrumentation grants (S10RR029668 and S10RR027303). We thank Mary West and the CIRM/QB3 Shared Stem Cell Facility/High-Throughput Screening Facility for technical support, as well as Timothy Brown (Thermo Fisher Scientific) for flow cytometry support. We thank Daniel Toso and Paul Tobias for their assistance with cryo-EM data collection at the Bay Area Cryo-EM facility at the University of California, Berkeley. We also thank Emeric Charles, Rob Nichols, Luke Oltrogge, Avi Flamholz, and Joshua Cofsky for technical support and productive discussions.

## Author contributions

A.S. and S.A.H. contributed equally to the work presented in the manuscript. S.A.H., A.S., C.F., T.G.L., B.L.O., and D.F.S. conceived and planned experiments. S.A.H., A.S., C.F., T.G.L., R.L., S.K., and M.A. performed experiments. S.A.H., A.S., C.F., T.G.L., and M.L. analyzed the data. B.L.O., B.T.S., J.A.D., and D.F.S. provided material and conceptual support. S.A.H., A.S., C.F., T.G.L., and D.F.S. wrote the manuscript. All authors reviewed the final manuscript.

## Competing interests

UC Regents have filed a patent related to this work; application number WO2020005980A1. J.A.D., D.F.S., S.A.H., and B.L.O. are listed as inventors. S.A.H. is an employee of Scribe Therapeutics. B.L.O. and B.T.S. are co-founders and employees of Scribe Therapeutics. C.F. is a co-founder of Mirimus, Inc. J.A.D. is a co-founder of Caribou Biosciences, Editas Medicine, Intellia Therapeutics, Scribe Therapeutics, and Mammoth Biosciences. J.A.D. is a scientific advisory board member of Caribou Biosciences, Intellia Therapeutics, eFFECTOR Therapeutics, Scribe Therapeutics, Synthego, Metagenomi, Mammoth Biosciences, and Inari. J.A.D. is a member of the board of directors at Driver and Johnson & Johnson. D.F.S. is a co-founder of Scribe Therapeutics and a scientific advisory board member of Scribe Therapeutics and Mammoth Biosciences. All the other authors declare no competing interests.
