## [Peer Review File · Nature Communications]

Reviewers' Comments:

Reviewer #1:

Remarks to the Author:

In the article "Comprehensive deletion landscape of CRISPR-Cas9 identifies minimal RNA-guided DNA-binding modules" Shams and Higgins et al. have established a powerful and innovative new technique to reduce the size of multidomain proteins using minimization by iterative size-exclusion and recombination (MISER). MISER comprehensively assays all possible deletions of a protein, and in this study, Shams and Higgins et al. leverage MISER to produce a deletion landscape for nuclease-dead SpCas9 (dCas9). In so doing, the authors identify key new insights into the biology of Cas9 DNA binding, in particular novel structure-function relationships, and create smaller dCas9 deletion variants that retain DNA binding and are capable of transcriptional repression in bacteria and human cells. Importantly, the authors have demonstrated that the MISER technique is a versatile, comprehensive, and unbiased method to probe the deletion landscape of proteins, which will be broadly impactful to the scientific community. Although Shams and Higgins have only applied MISER to dCas9 herein, the dCas9 protein is the perfect testbed for MISER because dCas9 has a multi-domain architecture, defined high-throughput assays for DNA cutting and binding, and is widely used in basic and translational biomedical research. Overall, the manuscript meets the publication criteria of Nature Communications, with only a few minor revisions needed.

Comments:

1. Although not crucial, given that the study is focused on using the MISER strategy for nuclease-dead Cas9 (dCas9) activity, the authors could consider specifying this in the abstract/introduction to avoid ambiguity with conventional Cas9-based genome editing. Alternatively, given the widespread adoption of nuclease active Cas9 - applying MISER in the context of conventional Cas9-based genome editing (e.g. nuclease activity) could dramatically improve the impact of this study.
2. In Figure 2a, the authors have demonstrated the *in vivo* transcriptional repression activity of Δ REC2, Δ REC3, Δ HNH and Δ RuvC, whereas in the BLI assay (Figure 2c) Δ HNH and Δ RuvC data is missing. If the authors have BLI data for Δ HNH and Δ RuvC mutants, it would be helpful if that data were included in this Figure.
3. In Figure 2d, CRISPRi activity was performed in mammalian U-251 cells at the PCNA locus, permitting the authors to conclude that Δ REC2 and Δ REC3 mutants lose their repressive activity at day 5, whereas Δ HNH and Δ RuvC mutants maintain repressive capacity. Given the nuances of site/locus-specific gene regulation, and the *n* of 2, the authors should include at least one more locus to confirm and strengthen their conclusion.
4. Many of the conclusions resulting from this study come from either *in vitro* DNA binding activity or *in vivo* repression (GFP/RFP in bacteria and PCNA in mammalian cells). What is the rationale for choosing repression (CRISPRi) over activation (CRISPRa) after screening? CRISPRa activity can be much more robust and sensitive (upregulation of a gene could be up to thousands of fold). If the authors perform one or two CRISPRa assays using their Δ dCas9 variants, that could add substantial impact to the manuscript and readers.

Minor Comments:

5. In Figures 2d and 3d the authors should make it clear that KRAB is included beyond mention in the legend. For instance, the authors could consider adding KRAB to the x-axes of Figures 2d and 3d.
6. Page-9, line 247, "when targeted to RFP, Δ REC3 and Δ REC3 show less robust activity". Here one "REC3" should be "REC2".

Reviewer #2:

Remarks to the Author:

The authors present a clever method for creating large, near-comprehensive deletion libraries in

proteins. They apply the technique to studying small-size and whole domain deletion mutants in the multi-domain protein dCas9 and discovered functional dCas9 variants of less than 1000 amino acids in length. While other groups have generated truncated dCas9 mutants through rational design (for example by Ma et al, ref 19), Shams et al.'s method provides a general and effective way of creating a broad array of in-frame deletions, which can then be combined to miniaturize the protein further. The potential for using this technique to reduce immunogenicity is intriguing and should be a fruitful avenue of future research. We provide more specific comments below.

1. We think it best if the abstract said the study was of dCas9 (not Cas9) since many of the deletions discovered would inactivate the nuclease activity of Cas9
2. In the abstract, the authors say that MISER "comprehensively assays all possible deletions of a protein." We think that "assays" is the wrong word. MISER is a method to create libraries of deletions, not assay them. Also, the amount of white in Figure 1C indicates that a not insignificant portion of the deletion space was not observed in the deep sequencing. Thus, we think that alternate phrasing such as "MISER is designed to create all possible deletions of a protein" or "...designed to comprehensively create..." would be more appropriate.
3. Figure 1C shows that deletions that start or stop at certain parts of the gene are almost completely absent from the sequencing data. As an example, this can clearly be seen for deletions in the middle of the REC3 domain. We think that the authors should reference this apparent bias and provide an explanation for why this might be the case. For example, the scarcity of deletions in the REC3 domain seems to extend to most deletions starting or ending the middle of the REC3 domain, as evidence by the diagonal white stripes in Figure 1C that intersect in the center of the REC3. This suggests that the reason deletions are not often seen in this region is because insertions of the NheI and SpeI restriction sites were not as successful in this region. Because the paper is introducing MISER as a new method, discussion of biases observed in the library and their causes (and possible solutions) would be helpful to potential users of the method. In light of this, the authors might want to temper statements such as saying their method "comprehensively assays all possible deletions of a protein" or that it is "unbiased", while still conveying that they are able to construct a large portion of the possible deletions (and that the method is designed to create a comprehensive library).
4. The sublibraries have a lot of library members well outside the expected deletion range as indicated by Fig 1C and FigS3A-D. Do the authors know the origin of these library members with large deletions? Do they arise because dCas9 is slightly toxic to bacteria, and large deletions offer a growth advantage (i.e. the dCas9 gene is prone to deletions in E. coli under these experimental conditions)? We think that the authors should point out the unexpected presence of large deletions in sublibraries 4 and 5 and at least speculate on the cause.
5. Line 209. "A similar phenomenon is observed with the Δ REC3 variant, the binding defect is less pronounced than in Δ REC2". Figure 2C shows the opposite, the binding defect with Δ REC3 is more pronounced than with Δ REC2.
6. Fig. 3C. We understand that binding of the multi-deletion dCas9s were poor at 300 nM, and thus 1000 nM was used. Why wasn't dCas9 tested at 1000 nM to allow an equal comparison?
7. The legend of Fig S10B does not indicate the concentration of dCas9 (the black line). We presume it is 300 nM in both plots. But if that is the case, why is the dCas9 curve for the adjacent-bubble target well below the Δ 3CE and Δ 4CE curves in Fig S10B but just a little bit above in Fig 3C. Is the concentration of dCas9 in the right graph of Fig 3C actually 1000 nM and not 300 nM? (answering our previous question).
8. Line 294 the authors state that "the PAM-interrogation ability of the two constructs appeared to be intact, as evidenced by the sharp drop-off in signal during the dissociation phase". If we are like typical readers, an additional explanation would be helpful. We notice that in Figure S10, the drop-off during dissociation for a construct containing a PAM but no spacer is very sharp. But we are unsure if this is part of the authors rationale or not, as it would seem that BLI results using a construct with a spacer but lacking a PAM would be necessary to make this claim.
9. It is unfortunate that the Δ 3CE protein had reduced activity compared to the single deletion mutants when used in mammalian cells even though they were able to repress gene expression at a comparable strength to single deletion mutants in bacteria. Do the authors have a hypothesis for this difference?

Minor Points:

1. Lines 56-62. Another advantage of MISER that might be pointed out is that the deletions are

limited to those that are in-frame, whereas some of the other methods are not.

2. Figure 1C: We suggest it might help to explicitly state in the legend that this map is a composite of the slice 4 and slice 5 sublibrary deletions (i.e. Fig S3A and S3B).

3. Line 388-394. Is there an estimate of the concentration of dCas9 inside bacterial cells? Could the reason for better performance in vivo be that the in vivo concentration of dCas9 is higher than that tested in vitro?

Point-by-point Response to Reviewers

Comprehensive deletion landscape of CRISPR-Cas9 identifies minimal RNA-guided DNA-binding modules.

Shams and Higgins et. al. 2020. (Reviewer comments in black, authors' response in blue)

Reviewer #1

In the article "Comprehensive deletion landscape of CRISPR-Cas9 identifies minimal RNA-guided DNA-binding modules" Shams and Higgins et al. have established a powerful and innovative new technique to reduce the size of multidomain proteins using minimization by iterative size-exclusion and recombination (MISER). MISER comprehensively assays all possible deletions of a protein, and in this study, Shams and Higgins et al. leverage MISER to produce a deletion landscape for nuclease-dead SpCas9 (dCas9). In so doing, the authors identify key new insights into the biology of Cas9 DNA binding, in particular novel structure-function relationships, and create smaller dCas9 deletion variants that retain DNA binding and are capable of transcriptional repression in bacteria and human cells. Importantly, the authors have demonstrated that the MISER technique is a versatile, comprehensive, and unbiased method to probe the deletion landscape of proteins, which will be broadly impactful to the scientific community. Although Shams and Higgins have only applied MISER to dCas9 herein, the dCas9 protein is the perfect testbed for MISER because dCas9 has a multi-domain architecture, defined high-throughput assays for DNA cutting and binding, and is widely used in basic and translational biomedical research. Overall, the manuscript meets the publication criteria of Nature Communications, with only a few minor revisions needed.

Comments:

1. Although not crucial, given that the study is focused on using the MISER strategy for nuclease-dead Cas9 (dCas9) activity, the authors could consider specifying this in the abstract/introduction to avoid ambiguity with conventional Cas9-based genome editing. Alternatively, given the widespread adoption of nuclease active Cas9 - applying MISER in the context of conventional Cas9-based genome editing (e.g. nuclease activity) could dramatically improve the impact of this study.

We have changed the abstract per the suggestion. We agree this would be highly interesting data to add and are working to include active cutting in a follow-up study.

2. In Figure 2a, the authors have demonstrated the in vivo transcriptional repression activity of Δ REC2, Δ REC3, Δ HNH and Δ RuvC, whereas in the BLI assay (Figure 2c) Δ HNH and Δ RuvC data is missing. If the authors have BLI data for Δ HNH and Δ RuvC mutants, it would be helpful if that data were included in this Figure.

We have added BLI data for the Δ HNH and Δ RuvC constructs in Fig. 2C.

3. In Figure 2d, CRISPRi activity was performed in mammalian U-251 cells at the PCNA locus, permitting the authors to conclude that Δ REC2 and Δ REC3 mutants lose their repressive activity at day 5, whereas Δ HNH and Δ RuvC mutants maintain repressive capacity. Given the nuances of site/locus-specific gene regulation, and the n of 2, the authors should include at least one more locus to confirm and strengthen their conclusion.

We have performed a CRISPRi experiment to test the function of the MISER constructs in U-251 cells. We believe that this shows a better demonstration of the constructs in a mammalian system and addresses the reviewer's concerns.

Briefly, we used KRAB CRISPRi fusion constructs to repress two essential genes in U-251 cells, PCNA and RPA1, in competitive proliferation assays. MISER-expressing cells were then transduced with constructs expressing either non-targeting sgRNA or one of several possible sgRNAs targeting PCNA/RPA1 and an additional mCherry reporter). The cells were then mixed with the parental population and monitored via flow-cytometry over several days. A reduction in the mCherry+ signal indicated repression of the essential genes, resulting in cell death and a decrease of the mCherry+ ratio. The data suggests that the Δ REC2, Δ HNH, and Δ RuvC single-deletion constructs have repression activity similar to dCas9 for all guides tested, except for sgRNA-i8 targeting RPA1. Δ REC3 exhibited weak function, which could result from the significant level of repression required to observe the growth phenotype, but the previous RT-qPCR data confirms it is indeed functional. These data are now included in Figures 2E and 3E with relevant changes to the manuscript.

4. Many of the conclusions resulting from this study come from either in vitro DNA binding activity or in vivo repression (GFP/RFP in bacteria and PCNA in mammalian cells). What is the rationale for choosing repression (CRISPRi) over activation (CRISPRa) after screening? CRISPRa activity can be much more robust and sensitive (upregulation of a gene could be up to thousands of fold). If the authors perform one or two CRISPRa assays using their Δ dCas9 variants, that could add substantial impact to the manuscript and readers.

We agree that testing the MISER constructs in multiple paradigms would be a nice addition! We are presenting the CRISPRi experiments because it is perhaps the most commonly employed CRISPR fusion effector strategy and generally works more consistently than CRISPRa. We believe that the main focus of this study is demonstration of the MISER technique rather than development of any single functional Cas9 variant, but, of course, want to show versatility in the long run and are pursuing such experiments.

Minor Comments:

1. In Figures 2d and 3d the authors should make it clear that KRAB is included beyond mention in the legend. For instance, the authors could consider adding KRAB to the x-axes of Figures 2d and 3d.

Added "KRAB" to the construct names in the x-axes of Figures 2D and 3D.

2. Page-9, line 247, “when targeted to RFP, Δ REC3 and Δ REC3 show less robust activity”. Here one “REC3” should be “REC2”.

Fixed.

Reviewer #2

The authors present a clever method for creating large, near-comprehensive deletion libraries in proteins. They apply the technique to studying small-size and whole domain deletion mutants in the multi-domain protein dCas9 and discovered functional dCas9 variants of less than 1000 amino acids in length. While other groups have generated truncated dCas9 mutants through rational design (for example by Ma et al, ref 19), Shams et al.’s method provides a general and effective way of creating a broad array of in-frame deletions, which can then be combined to miniaturize the protein further. The potential for using this technique to reduce immunogenicity is intriguing and should be a fruitful avenue of future research. We provide more specific comments below.

1. We think it best if the abstract said the study was of dCas9 (not Cas9) since many of the deletions discovered would inactivate the nuclease activity of Cas9

We have clarified in the Abstract the studies were performed on dCas9 specifically.

2. In the abstract, the authors say that MISER “comprehensively assays all possible deletions of a protein.” We think that “assays” is the wrong word. MISER is a method to create libraries of deletions, not assay them. Also, the amount of white in Figure 1C indicates that a not insignificant portion of the deletion space was not observed in the deep sequencing. Thus, we think that alternate phrasing such as “MISER is designed to create all possible deletions of a protein” or “...designed to comprehensively create...” would be more appropriate.

We have clarified in the Abstract that the MISER technique “makes” rather than assays all deletions in a gene.

3. Figure 1C shows that deletions that start or stop at certain parts of the gene are almost completely absent from the sequencing data. As an example, this can clearly be seen for deletions in the middle of the REC3 domain. We think that the authors should reference this apparent bias and provide an explanation for why this might be the case. For example, the scarcity of deletions in the REC3 domain seems to extend to most deletions starting or ending the middle of the REC3 domain, as evidence by the diagonal white stripes in Figure 1C that intersect in the center of the REC3. This suggests that the reason deletions are not often seen in this region is because insertions of the NheI and SpeI restriction sites were not as successful in this region. Because the paper is introducing MISER as a new method, discussion of biases observed in the library and their causes (and possible solutions) would be helpful to potential users of the method. In light of this, the authors might want to temper statements such as saying their method “comprehensively assays all possible deletions of a

protein” or that it is “unbiased”, while still conveying that they are able to construct a large portion of the possible deletions (and that the method is designed to create a comprehensive library).

It is true that there are gaps across certain regions where we do not have data, and we find it likely to be due to inefficient *SpeI* or *NheI* insertion in the recombineering step as the reviewer suggests. This sequence-dependent bias in recombineering efficiency has been observed previously, and we can address this drawback in the manuscript. Despite the unpredictable nature of these low mutagenesis sequences, it has been shown that subsequent mutagenesis can account for these biases and produce more evenly distributed libraries (Higgins, Ouonkap, and Savage, *ACS Synth Biol*, 2017).

4. The sublibraries have a lot of library members well outside the expected deletion range as indicated by Fig 1C and FigS3A-D. Do the authors know the origin of these library members with large deletions? Do they arise because dCas9 is slightly toxic to bacteria, and large deletions offer a growth advantage (i.e. the dCas9 gene is prone to deletions in *E. coli* under these experimental conditions)? We think that the authors should point out the unexpected presence of large deletions in sublibraries 4 and 5 and at least speculate on the cause.

The variants on the right in Figures S3 C & D are products of ligations with large dCas9 deletions that are present in the gel slices themselves, prior to any screening or selection. As the reviewer suggests, this is likely due to the fact that smaller plasmids transform better, have potentially less genetic burden, and do not express Cas9, which can be toxic. Thus, smaller plasmids containing very little of the dCas9 sequence probably allow a greater number of large deletions to proliferate in the cell population. We have now explicitly noted this in the Deep Sequencing section of the Supplementary Information.

5. Line 209. “A similar phenomenon is observed with the Δ REC3 variant, the binding defect is less pronounced than in Δ REC2”. Figure 2C shows the opposite, the binding defect with Δ REC3 is more pronounced than with Δ REC2.

We thank the reviewer for pointing this out! It was the result of a figure formatting error. The figure was incorrectly labeled and has now been corrected. The red traces belong to Δ REC2 and the blue traces belong to Δ REC3.

6. Fig. 3C. We understand that binding of the multi-deletion dCas9s were poor at 300 nM, and thus 1000 nM was used. Why wasn't dCas9 tested at 1000 nM to allow an equal comparison?

We have tried the experiment with dCas9 at 1000 nM, but unfortunately the binding signal oversaturated the BLI sensors and the data was not meaningful. We have therefore chosen not to include that data.

7. The legend of Fig S10B does not indicate the concentration of dCas9 (the black line). We presume it is 300 nM in both plots. But if that is the case, why is the dCas9 curve for the adjacent-bubble target well below the $\Delta 3CE$ and $\Delta 4CE$ curves in Fig S10B but just a little bit above in Fig 3C. Is the concentration of dCas9 in the right graph of Fig 3C actually 1000 nM and not 300 nM? (answering our previous question).

The BLI traces were incorrectly normalized to individual maxima in Figure 3, but normalized to the maximum value in the dCas9 trace in Figure S10, (with $y=1.00$ representing the maximum value of [300 nM dCas9]). The traces in Figure 3 have now been normalized to the [300 nM dCas9] maximum. We thank the reviewer for pointing out this error.

8. Line 294 the authors state that “the PAM-interrogation ability of the two constructs appeared to be intact, as evidenced by the sharp drop-off in signal during the dissociation phase”. If we are like typical readers, an additional explanation would be helpful. We notice that in Figure S10, the drop-off during dissociation for a construct containing a PAM but no spacer is very sharp. But we are unsure if this is part of the authors rationale or not, as it would seem that BLI results using a construct with a spacer but lacking a PAM would be necessary to make this claim.

In Fig. S10 B we have added BLI data for the $\Delta 3CE$ and $\Delta 4CE$ constructs as they interact with a dsDNA target that does not have any complementarity with the sgRNA (i.e. no spacer) but still contains a NGG PAM. These traces show increased binding signal during the association phases for the triple- and quadruple-deletion constructs, but that signal immediately drops off to near-baseline levels during the dissociation phases. This suggests that the interaction was transient, and most likely due to the PAM surveillance activity still retained by the $\Delta 3CE$ and $\Delta 4CE$ RNPs.

9. It is unfortunate that the $\Delta 3CE$ protein had reduced activity compared to the single deletion mutants when used in mammalian cells even though they were able to repress gene expression at a comparable strength to single deletion mutants in bacteria. Do the authors have a hypothesis for this difference?

At this time we can only speculate that the $\Delta 3CE$ has some overall kinetic defect that renders it unable to access its dsDNA target in mammalian cells. It is possible that the loss of the two REC domains prevents the RNP from invading genomic DNA, especially in the context of mammalian cells where DNA is quite dynamic and under constant maintenance and rearrangement by other DNA binding proteins.

Minor Points:

1. Lines 56-62. Another advantage of MISER that might be pointed out is that the deletions are limited to those that are in-frame, whereas some of the other methods are not.

Added a line in the Introduction clarifying that MISER makes in-frame deletions in a gene.

2. Figure 1C: We suggest it might help to explicitly state in the legend that this map is a composite of the slice 4 and slice 5 sublibrary deletions (i.e. Fig S3A and S3B).

Explicitly noted that Figure 1C is a composite of the heatmaps in Figures S3A and S3B in the Figure 1 legend.

3. Line 388-394. Is there an estimate of the concentration of dCas9 inside bacterial cells? Could the reason for better performance *in vivo* be that the *in vivo* concentration of dCas9 is higher than that tested *in vitro*?

It is very difficult to quantify the total amount of dCas9 inside the bacterial cell over the course of the screening pipeline due to a number of reasons. Firstly, the dCas9 cassette can have leaky expression, which would confound any measurements made after induction. Secondly, the fluorescence measurements are made after allowing the fluorophores to mature for some time in 4°C, so we assume that the number of dCas9 molecules in the cell are at a saturating equilibrium, and the difference in fluorescence are only due to differences in dCas9 binding. We hesitate to speculate precisely why we see better performance *in vivo*, other than to suggest that the genomic DNA topology inside cells is more dynamic compared to *in vitro* (due to replication, transcription, recombination, etc.), and there are more frequent opportunities for a DNA-binding protein (although defective) to interrogate open or single-stranded sequences.

Reviewers' Comments:

Reviewer #1:

Remarks to the Author:

The authors have addressed all of my concerns. Their work in this manuscript will be very impactful and useful for the community. I feel that the revised manuscript is suitable for publication in Nature Communications.

Reviewer #2:

Remarks to the Author:

The authors have satisfactorily addressed our comments.